# Generating Commonsense Counterfactuals for Stable Relation Extraction

**Xin Miao[1], Yongqi Li[1], Tieyun Qian[1,2*]**
[1] School of Computer Science, Wuhan University, China
[2] Intellectual Computing Laboratory for Cultural Heritage, Wuhan University, China
{miaoxin,liyongqi,qty}@whu.edu.cn

## Abstract

Recent studies on counterfactual augmented data have achieved great success in the coarse-grained natural language processing tasks. However, existing methods encounter two major problems when dealing with the fine-grained relation extraction tasks. One is that they struggle to accurately identify causal terms under the invariant entity constraint. The other is that they ignore the commonsense constraint. To solve these problems, we propose a novel framework to generate commonsense counterfactuals for stable relation extraction. Specifically, to identify causal terms accurately, we introduce an intervention-based strategy and leverage a constituency parser for correction. To satisfy the commonsense constraint, we introduce the concept knowledge base WordNet and design a bottom-up relation expansion algorithm on it to uncover commonsense relations between entities. We conduct a series of comprehensive evaluations, including the low-resource, out-of-domain, and adversarial-attack settings. The results demonstrate that our framework significantly enhances the stability of base relation extraction models[1].

## 1 Introduction

The relation extraction (RE) task aims to extract the semantic relation between entities in the text. RE can facilitate a wide range of downstream applications such as knowledge graph construction (Gajendran et al., 2023) and question answering (Li et al., 2022b), having aroused much attention in recent years. Typical RE methods based on pre-trained language models like BERT (Devlin et al., 2018), or RoBERTa (Liu et al., 2019). While getting impressive performance, they exhibit a certain degree of instability in RE (Li et al., 2021; Chen et al., 2023), even the ChatGPT (Wei et al., 2023).

---

*Corresponding author.
[1]The code and data used in our experiments are available at: https://github.com/NLPWM-WHU/CCG

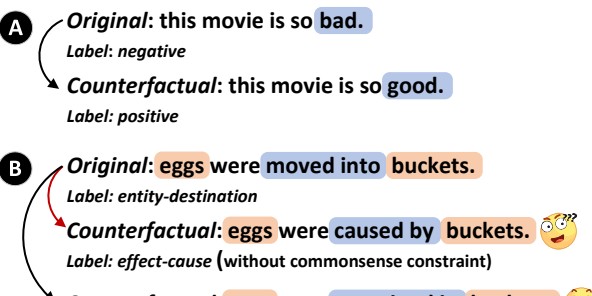

Figure 1: Counterfactuals in sentiment analysis (A) and relation extraction (B) tasks. The words in blue denote causal terms, and those in orange denote entities.

Instability has always been associated with neural networks. For example, Li et al. (2021) finds that BERT performs poorly in the de-biased set constructed by replacing high-frequency words with low-frequency ones. The instability is primarily caused by spurious correlations between the high-frequency words and labels. Specifically, this can be attributed to neural networks tending to learn shortcuts between instances and labels (Geirhos et al., 2020). Such property may hurt the model's stability in some challenging scenarios, e.g., low-resource or cross-domain scenarios.

In attempts to mitigate spurious correlations, counterfactual augmented data (CAD) is a rising trend. CAD can be defined as: flipping the label of an instance with minimal editing (Kaushik et al., 2019), which explicitly provides localized views of decision boundaries (Treviso et al., 2023). Current CAD methods consist of the following three steps: (1) identifying causal terms, that are causally related to the label; (2) editing the causal term, to flip the label to other prepended ones; and (3) filtering out inconsistent instances, where the predicted label is different from the prepended one.

Existing methods have made great success in coarse-grained natural language processing tasks like sentiment analysis (Yang et al., 2021; Chen

et al., 2021; Howard et al., 2022) and natural language inference (Wu et al., 2021; Ross et al., 2021; Wen et al., 2022). However, the fine-grained RE task is under-explored (Zhang et al., 2023). Due to the intrinsic property of RE, i.e. invariant entity constraint and commonsense constraint, existing methods encounter the following two challenges.

Firstly, existing methods struggle to accurately identify causal terms, due to the constraint of invariant entities (Zhang et al., 2023). That is, keeping the entities unaltered. The state-of-the-art methods primarily employ the model's gradient attributions to identify the causal term (Ross et al., 2021; Wen et al., 2022), i.e. selecting certain words with the highest gradient. However, entities convey more information (Zhang et al., 2023), which implies that the gradient of causal terms can be confounded by the related entities. Zhang et al. (2023) propose a syntactic-tailored strategy, i.e. taking the words along the shortest dependency path (SDP) between entities as causal terms. The drawback is its heavy reliance on syntactic parsing quality.

Secondly, and more importantly, all existing methods ignore the commonsense constraint. They assume that all labels can be treated as prepended ones. This is feasible in coarse-grained tasks. For example, in sentiment analysis, simply replacing the causal terms can flip the sentiment orientation, as shown in Fig. 1 (A). However, for RE, the nature of entities imposes commonsense constraints on causal terms, as shown in Fig. 1 (B). In brief, the relation between entities should conform to the cognitive understanding of the real world. Although existing methods pre-train a RE base model on the existing dataset for consistency filtering (Ross et al., 2021; Wen et al., 2022; Zhang et al., 2023), there remains an issue of entanglement with the base model. For example, in low-resource scenarios, the base model is severely affected by spurious correlations and propagates the errors backward.

To address these two challenges, we present a **C**ommonsense **C**ounterfactual **G**eneration (CCG) framework for stable relation extraction. Firstly, to identify the causal terms accurately, we present a novel intervention-based strategy, which can accurately identify the editable words by potential interventions. After obtaining preliminary results, we further utilize the constituency parser for correction. Secondly, to satisfy the commonsense constraint, we leverage the concept knowledge base WordNet (Miller, 1995) and utilize its hierarchical

structure knowledge i.e. the common hypernyms, to interconnect entities, thereby expanding their relations. We design a bottom-up relation expansion algorithm for implementation, which can uncover commonsense relations between entities.

We conduct comprehensive evaluations on a series of challenging scenarios, including the low-resource, out-of-domain, and adversarial-attack settings. The results demonstrate that our CCG generates more reasonable counterfactuals, which can consistently enhance RE models' stability in various scenarios. Furthermore, we also conduct a series of in-depth analysis studies. The results confirm the effectiveness of various CCG strategies and demonstrate that our generated counterfactuals are not only semantically readable but, more importantly, consistent with commonsense.

## 2 Related Work

### 2.1 Relation Extraction (RE)

Early deep learning RE models get promising performance by either extracting better semantic features from sentences (Zhou et al., 2016; Zhang et al., 2017) or incorporating syntactic features over dependency graph (Guo et al., 2019; Mandya et al., 2020). More recently, pre-trained language models (PLMs) (Devlin et al., 2018) based RE methods can achieve superior performance (Yamada et al., 2020; Qin et al., 2021; Chen et al., 2022).

Despite the remarkable progress, the stability of RE methods has been largely neglected from the perspective of CAD. The only research towards this direction (Zhang et al., 2023) ignores the commonsense constraint. Moreover, its performance is entangled with the base model due to the consistency filtering. In contrast, our CCG takes commonsense into account and is independent of the base model during the counterfactual generation process. This ensures that our counterfactuals are reasonable and not constrained by the base model.

### 2.2 Counterfactual Data Augmentation

Inspired by causal inference (Pearl, 2009; Pearl and Mackenzie, 2018), studies to improve model stability by eliminating spurious correlation features have received increasing attention (Kaushik et al., 2019; Niu et al., 2021). Among them, counterfactual data augmentation methods have become one of the mainstreams, especially in natural language understanding tasks (Kaushik et al., 2019; Garg and Ramakrishnan, 2020; Wang and Culotta,

2021; Ross et al., 2021; Robeer et al., 2021; Chen et al., 2021; Yang et al., 2021; Wen et al., 2022; Howard et al., 2022; Zhang et al., 2023). These methods generate augmented data by replacing the causal terms in original samples and flipping the corresponding labels. In this way, the impact of causal items on the output is emphasized while the importance of spurious correlation features is weakened (Kaushik et al., 2019; Joshi and He, 2021).

However, most of these approaches have been performed on coarse-grained natural language understanding tasks, such as sentence-level sentiment analysis (Kaushik et al., 2019; Chen et al., 2021; Yang et al., 2021; Dixit et al., 2022; Howard et al., 2022), text classification (Garg and Ramakrishnan, 2020; Wang and Culotta, 2021), and natural language inference (Wen et al., 2022; Dixit et al., 2022). For fine-grained tasks like relation extraction, one key property is the commonsense associated with the entity pair. If the commonsense constraint is violated, the obtained counterfactuals may negatively affect the model's performance and stability. Based on this observation, this work explicitly introduces commonsense knowledge into the counterfactual generation for the first time.

## 3 Methodology

### 3.1 Task Definition

Let $\mathcal{X} = \{(x_i = (e_i, c_i), y_i)\}$ be the dataset, where $x_i \in \mathcal{X}$ is a sentence contains a known entity pair $e_i$ and their context $c_i$. $y_i \in \mathcal{Y}$ is the corresponding relation of the entity pair. Given the sentence $x_i$, RE aims to extract the relation $y_i$.

For a given instance $(x_i = (e_i, c_i), y_i)$, we define the commonsense counterfactual augmented data (CCAD) as: generating a counterfactual instance $(\hat{x} = (e_i, \hat{c}_i), \hat{y}_i)$ that meets the following requirements. (1) ***Minimal perturbation:*** only the causal term in $\hat{c}_i$ is edited. (2) ***Commonsense constraint:*** $\hat{y}_i$ needs to satisfy the commonsense constraint regarding $e_i$ and $\hat{y}_i \neq y_i$. (3) ***Label flipping:*** $\hat{x}_i$ needs to be consistent with $\hat{y}_i$. Note that requirements (1) and (3) are inherited from previous works (Kaushik et al., 2019; Garg and Ramakrishnan, 2020; Wang and Culotta, 2021), and requirement (2) is proposed by this work.

Our framework CCG consists of three components: causal terms identification, relation expansion, and controllable editing. Each module satisfies one of the aforementioned requirements. The overview of CCG is shown in Figure 2.

### 3.2 Causal Terms Identification

To meet the requirement of *minimal perturbations*, the causal terms should be identified precisely. Otherwise, the errors will affect the later counterfactual generation. To this end, we propose an intervention-based strategy combined with a constituency parser. The details are shown in Figure 2 (1).

**MLM Intervenor** We first employ a masked language model (MLM) as the intervenor. It intervenes on each contextual word separately. Specifically, it masks each word individually using `<mask>` symbol and then performs cloze-style filling. To ensure the rationality of each intervention, we only select the top $\mathcal{N}$ words predicted by the intervenor.

**Trained Indicator** The indicator is a trained RE-based model based on the existing dataset. It predicts each intervention instance separately. If a predicted label changes, we regard the intervened word as the preliminary causal term. It is worth noting that even in low-resource scenarios, the base model can still work because the decision space for determining the change of predictions (binary classification) is much smaller than the decision space for predicting the actual outcomes (multi-class classification, directly proportional to the number of relations), which greatly reduces the complexity of the problem. The process can be formalized as:

$$\phi(w_i^j) = \begin{cases} 1, & if \ b(x_i) \neq b(x_i \backslash w_i^j; r(w_i^j)) \\ 0, & otherwise, \end{cases} \quad (1)$$

where $r(w_i^j)$ denotes the intervention word, $x_i \backslash w_i^j$ represents the sentence $x_i$ without the word $w_i^j$, and $b$ is the trained indicator.

**Constituency Parsing based Causal Item Identification** After obtaining the preliminary causal term, We observed that words belonging to the same phrase as the preliminary causal term were not identified, although they should be treated as a whole. To alleviate this issue, we utilize a constituency parse tree to find the overlooked causal term. To achieve automatic operation, we design the processing workflow: (1) traverse upward from the node where the preliminary causal term are located to find the nearest verb phrase (NVP) or prepositional phrase (NPP). (2) If NVP is found, traverse its child nodes and label the verb (VP) or preposition (IN) nodes as causal items, then finish. (3) If NPP is found, traverse its child nodes and label the preposition (IN) nodes as causal items.

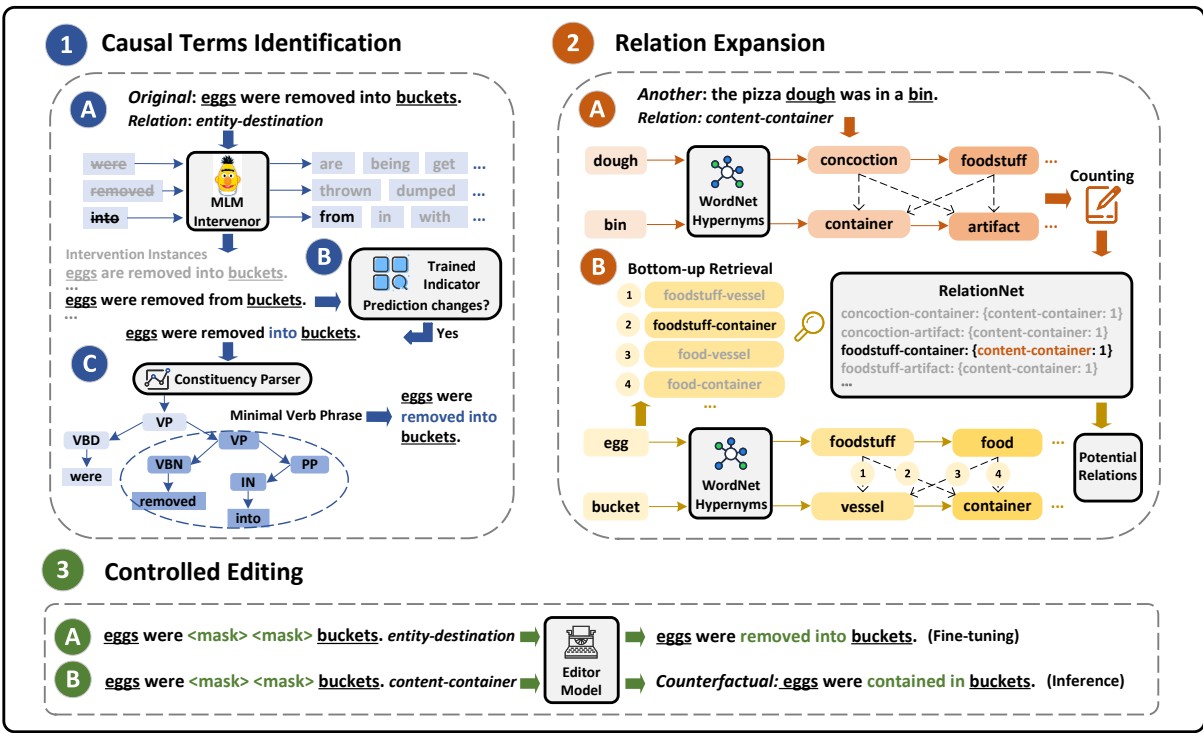

Figure 2: An overview of CCG framework, which consists of three modules, i.e., causal term identification, relation expansion, and controlled editing. The numbers 1, 2, and 3 in the colored circle denote the module id.

## 3.3 Commonsense Constrained Relation Expansion

To meet the requirement of *commonsense constraint*, we incorporate a concept knowledge base WordNet to expand the relations among entities. Specifically, to achieve relation transitivity, it is necessary to establish connections between entities. Hypernymy (super-name) are transitive relations between concepts (Miller, 1995), e.g., "primate" is a hypernym for "human". This semantic relation organizes the meanings of nouns into a hierarchical structure. Moreover, the concepts gradually become broader in the hierarchy of hypernyms. Based on this property, entities can be connected through common hypernyms. Furthermore, we assume that for any entity pair, the lower common hypernym in the hierarchical structure, the closer their semantics are, and the more likely their relations can be expanded. We implemented this idea through Module 2, as shown in Figure 2 (2).

**RelationNet Construction** Given a known instance, we first obtain the hypernyms of entities through WordNet. Then, we pairwise match the hypernyms between entities to form hypernym pairs. Finally, we respectively use the hypernym pairs as keys in RelationNet, while the current relation and

its occurrence count are accumulatively recorded as the corresponding values. The entire process can be formalized in lines 1-9 of Algorithm 1.

**Bottom-up Retrieval** We also obtain the hypernyms of entities through WordNet. Then, we pairwise match the hyperyms between entities in a bottom-up strategy. That is, priority is given to the lowest-level hypernym pair, and then proceeding using the similar strategy iteratively. According to the matching order, each hypernym pair is used to retrieve records from RelationNet. If a record exists and contains a new relation, we add it to the potential relations list. Due to the lower common hypernym in the hierarchical structure, the closer their semantics, our bottom-up retrieval strategy can proactively identify semantically closer entity pairs with different relations. As a result, the relations discovered earlier come from the entity pairs with more similar semantics, hence it should be given a higher priority. We select the top $\mathcal{K}$ relations for experimentation. A more detailed process is described in lines 10-25 in Algorithm 1. Note that the ratio threshold $\mathcal{H}$ can be adjusted to control the scope of upward retrieval. A smaller $\mathcal{H}$ will yield more accurate relations but less data, while a larger $\mathcal{H}$ will have the opposite effect. We will discuss the impacts of these parameters in §5.8.

**Algorithm 1:** Bottom-up Relation Expansion.

**Require:** Train set $\mathcal{X} = \{(x_i = (e_i^1, e_i^2), y_i)\}$, WordNet $\mathcal{W}$, Input instance $(x = (e_1, e_2), y)$, Ratio threshold $\mathcal{H}$, Top $\mathcal{K} = n$;

**Output:** Potential relations $\hat{\mathcal{Y}} = \{\hat{y_1}, \hat{y_2}, ...\hat{y_n}\}$, RelationNet $\mathcal{R}$.

```
 1: for (e_i^1, e_i^2) in X do
 2:     H_i^1 = W(e_i^1);
 3:     H_i^2 = W(e_i^2);
 4:     for h_i^1 in H_i^1 do
 5:         for h_i^2 in H_i^2 do
 6:             R = R ∪ {(h_i^1, h_i^2) : {y_i: 1}};
 7:         end for
 8:     end for
 9: end for
10: H_1 = W(e_1);
11: H_2 = W(e_2);
12: for hop in range(len(H_1 + H_2) * H) do
13:     H = {(hop_1, hop_2)|(hop_1 + hop_2 =
        hop) ∧ (hop_1 ≥ 0) ∧ (hop_2 ≥ 0)};
14:     for (hop_1, hop_2) in H do
15:         h_1 = H_1[hop_1];
16:         h_2 = H_2[hop_2];
17:         Ŷ_hop = Ŷ_hop ∪ {R[h_1-h_2]};
18:     end for
19:     sort Ŷ_hop;
20:     for ŷ_hop in Ŷ_hop do
21:         if ŷ_hop ≠ y and ŷ_hop not in Ŷ then
22:             Ŷ = Ŷ ∪ {ŷ_hop};
23:         end if
24:     end for
25: end for
```

## 3.4 Controlled Editing

To meet the requirement of *label flipping*, we need to edit the causal term conditioned on the uncovered new relation. The process can be regarded as controlled editing. We employ a prompt learning (Carlsson et al., 2022; Yang et al., 2022) method to accomplish this goal. Our proposed method consists of two stages, including fine-tuning and inference, as shown in Figure 2 (3).

At the fine-tuning stage, we put the available data into the pre-defined template and use them to fine-tune a generative model. Then, at the inference stage, we input the prompt that contains the new relation to the fine-tuned generative model and let it generate the content. Note the new relation is just the prompt, and the goal is to generate the masked tokens for satisfying this prompt. Finally, we treat the generated content as the counterfactual.

# 4 Evaluation Protocol

## 4.1 Low-resource Settings

Spurious correlations are particularly prevalent in low resource settings (Nan et al., 2021). To validate that CCG can mitigate such impact, we conduct

experiments on the low-resource SemEval (Hendrickx et al., 2019) under two scenarios i.e. 1%, 3%, 5%, 10% (Li et al., 2022a) and 2-shot, 4-shot, 8-shot, 16-shot, 32-shot (Chen et al., 2022).

## 4.2 Out-of-domain Setting

Spurious correlations also exist in out-of-domain settings, due to domain-specific tokens (Calderon et al., 2022). To validate CCG's performance in this scenario, following previous work (Zhang et al., 2023), we employ the ACE 2005 dataset (Grishman et al., 2005) for evaluation. We select four sub-datasets from different domains, including weblogs (WL), conversation (BC), broadcast news (BN), and newswire (NW). Specifically, we conduct experiments with WL→BC, WL→BN and WL→NW settings, respectively. Taking the WL→BC setting as an example, we use WL as the training set, and BC as the test set.

## 4.3 Adversarial-attack Settings

Adversarial attacks are a common way for evaluating model robustness (Lin et al., 2021). We construct an adversarial-attack test dataset based on the original test set of SemEval, namely RE-Attack[2]. In general, we employ a semi-automatic approach to intervene on causal terms and entities separately, generating perturbed instances with flipped and invariant labels. Table 7 shows the detailed statistics of the datasets used in our experiments.

## 4.4 Baselines

The comparative methods include the following three types. The first type, conventional methods: (1) **Synonym Replacement** (Zhang et al., 2015) replaces words with synonyms; (2) **Back Translation** (Sennrich et al., 2015) translates text into another language, then translates it back; (3) **BERT-MLM** (Jiao et al., 2019) replaces words by MLM.

The second type, counterfactual methods: (1) **MICE** (Ross et al., 2021) identifies causal terms by gradients and trains an editor to edit them; (2) **AutoCAD** (Wen et al., 2022) is similar with MICE, but it introduces unlikelihood strategy for the editor; (3) **CoCo** (Zhang et al., 2023) exploits syntactic and semantic dependency graphs to discover substitution. They all require consistency filtering.

The third type is LLM, we use **ChatGPT**, the most powerful LLM. We carefully designed a prompt to guide its generation of counterfactuals.

---

[2]To the best of our knowledge, RE-Attack is the first dataset for adversarial-attack testing in RE.

| Method | 1% | 3% | 5% | 10% | 2-shot | 4-shot | 8-shot | 16-shot | 32-shot |
|---|---|---|---|---|---|---|---|---|---|
| | | | | | **R-BERT** | | | | |
| Original | $33.26_{1.43}$ | $59.31_{1.46}$ | $68.66_{1.77}$ | $76.47_{1.14}$ | $23.48_{1.78}$ | $34.65_{2.32}$ | $50.26_{1.18}$ | $63.19_{2.93}$ | $75.58_{0.61}$ |
| Synonym Rep. | $29.78_{2.63}$ | $60.06_{1.15}$ | $69.57_{1.80}$ | $\underline{77.94}_{1.76}$ | $23.48_{0.70}$ | $35.78_{2.01}$ | $51.17_{1.12}$ | $64.37_{2.38}$ | $76.03_{0.57}$ |
| Back Trans. | $27.86_{2.21}$ | $56.68_{1.69}$ | $64.02_{2.28}$ | $75.53_{1.53}$ | $22.51_{1.21}$ | $28.36_{2.00}$ | $43.30_{0.63}$ | $61.80_{2.54}$ | $74.38_{1.48}$ |
| BERT-MLM | $36.46_{3.19}$ | $62.00_{2.38}$ | $67.97_{1.51}$ | $77.24_{0.90}$ | $26.09_{0.99}$ | $33.46_{3.06}$ | $48.54_{2.33}$ | $60.66_{1.98}$ | $75.06_{0.37}$ |
| MICE | - | $62.64_{0.05}$ | $\underline{72.51}_{0.61}$ | $77.02_{0.69}$ | - | - | $50.04_{1.23}$ | $\underline{69.92}_{0.87}$ | $75.78_{0.51}$ |
| AutoCAD | - | $\underline{62.78}_{1.32}$ | $71.81_{1.54}$ | $77.10_{0.40}$ | - | - | $50.41_{2.09}$ | $\underline{70.23}_{1.20}$ | $75.69_{0.86}$ |
| CoCo | - | $62.24_{1.10}$ | $69.97_{1.61}$ | $77.40_{0.66}$ | - | - | $49.05_{1.05}$ | $65.65_{1.98}$ | $\underline{76.38}_{0.27}$ |
| ChatGPT | $\underline{38.78}_{2.71}$ | $61.84_{1.23}$ | $67.90_{2.14}$ | $75.15_{1.10}$ | $\underline{26.29}_{1.52}$ | $\underline{36.73}_{1.94}$ | $54.31_{2.71}$ | $63.19_{1.17}$ | $72.69_{1.10}$ |
| **CCG** | $\mathbf{42.66}_{2.00}$ | $\mathbf{66.30}_{1.07}$ | $\mathbf{73.01}_{0.50}$ | $\mathbf{78.08}_{0.60}$ | $\mathbf{35.11}_{0.66}$ | $\mathbf{47.52}_{3.05}$ | $\mathbf{61.87}_{2.18}$ | $69.12_{0.94}$ | $\mathbf{76.88}_{0.71}$ |
| | | | | | **R-RoBERTa** | | | | |
| Original | $35.77_{2.41}$ | $64.27_{3.20}$ | $69.99_{1.84}$ | $78.27_{1.07}$ | $\underline{30.26}_{2.15}$ | $\underline{40.73}_{3.29}$ | $55.06_{3.96}$ | $66.22_{2.50}$ | $76.68_{0.48}$ |
| Synonym Rep. | $31.36_{4.63}$ | $62.89_{3.38}$ | $71.24_{3.73}$ | $\underline{78.51}_{0.74}$ | $27.18_{1.76}$ | $37.17_{4.19}$ | $55.09_{4.51}$ | $67.71_{2.93}$ | $76.14_{0.69}$ |
| Back Trans. | $29.79_{2.55}$ | $61.10_{2.85}$ | $68.00_{2.79}$ | $78.23_{1.13}$ | $26.81_{3.31}$ | $30.23_{3.47}$ | $50.93_{5.03}$ | $66.28_{1.23}$ | $76.23_{0.51}$ |
| BERT-MLM | $38.24_{3.99}$ | $63.90_{3.69}$ | $70.28_{3.06}$ | $77.65_{1.43}$ | $30.03_{2.97}$ | $35.24_{4.05}$ | $53.37_{5.42}$ | $64.50_{2.45}$ | $75.78_{0.70}$ |
| MICE | - | $\underline{68.42}_{1.96}$ | $\underline{74.76}_{1.69}$ | $78.21_{0.71}$ | - | - | $54.03_{4.38}$ | $70.77_{1.52}$ | $76.20_{0.67}$ |
| AutoCAD | - | $67.51_{2.18}$ | $74.74_{0.79}$ | $78.27_{0.79}$ | - | - | $54.11_{5.52}$ | $\underline{71.62}_{0.75}$ | $76.81_{0.63}$ |
| CoCo | - | $65.57_{2.73}$ | $74.16_{2.16}$ | $78.40_{0.88}$ | - | - | $54.08_{3.95}$ | $67.78_{0.93}$ | $\underline{77.20}_{1.31}$ |
| ChatGPT | $\underline{38.71}_{2.11}$ | $64.44_{1.34}$ | $70.14_{2.11}$ | $76.25_{0.52}$ | $27.00_{2.47}$ | $39.87_{4.14}$ | $\underline{56.34}_{2.68}$ | $65.25_{1.83}$ | $73.95_{0.86}$ |
| **CCG** | $\mathbf{44.05}_{3.73}$ | $\mathbf{70.16}_{1.18}$ | $\mathbf{75.32}_{0.70}$ | $\mathbf{79.29}_{1.11}$ | $\mathbf{39.86}_{1.89}$ | $\mathbf{49.53}_{2.04}$ | $\mathbf{61.64}_{3.82}$ | $\mathbf{72.35}_{0.89}$ | $\mathbf{77.63}_{1.50}$ |

Table 1: Results for low-resource settings on SemEval in terms of F1-scores. Scores in **bold** indicate the best result, and the underlined ones are the second best. The subscript denotes the standard deviation. "-" indicates there is no augmented data due to the base model being undertrained and not performing proper consistency filtering.

| Method | WL → BC | WL → BN | WL → NW | WL → BC | WL → BN | WL → NW |
|---|---|---|---|---|---|---|
| | | **R-BERT** | | | **R-RoBERTa** | |
| Original | $70.43_{2.45}$ | $70.55_{2.51}$ | $69.42_{1.41}$ | $74.17_{0.70}$ | $70.54_{0.87}$ | $74.93_{0.74}$ |
| Synonym Rep. | $71.77_{1.23}$ | $72.76_{0.72}$ | $70.38_{1.23}$ | $74.48_{0.80}$ | $\underline{71.75}_{0.80}$ | $74.43_{1.06}$ |
| Back Trans. | $71.76_{2.13}$ | $\underline{72.82}_{1.74}$ | $70.25_{1.26}$ | $\underline{74.95}_{1.23}$ | $71.64_{1.09}$ | $\underline{75.28}_{0.69}$ |
| BERT-MLM | $\underline{71.88}_{1.46}$ | $71.68_{0.81}$ | $\underline{70.49}_{1.00}$ | $73.54_{1.14}$ | $69.75_{0.30}$ | $73.42_{0.43}$ |
| MICE | $70.42_{1.60}$ | $70.72_{1.32}$ | $69.51_{1.06}$ | $73.53_{1.33}$ | $70.35_{0.73}$ | $73.86_{1.02}$ |
| AutoCAD | $70.76_{1.14}$ | $71.97_{1.34}$ | $69.97_{0.77}$ | $74.32_{1.04}$ | $71.23_{0.72}$ | $73.80_{0.52}$ |
| CoCo | $69.95_{1.14}$ | $71.17_{1.25}$ | $69.03_{1.12}$ | $73.79_{1.01}$ | $70.00_{0.48}$ | $73.56_{1.10}$ |
| ChatGPT | $52.70_{0.99}$ | $55.94_{1.21}$ | $54.51_{0.63}$ | $59.55_{0.50}$ | $60.11_{0.89}$ | $61.74_{1.13}$ |
| **CCG** | $\mathbf{73.16}_{0.60}$ | $\mathbf{73.11}_{0.78}$ | $\mathbf{70.69}_{0.91}$ | $\mathbf{75.55}_{0.75}$ | $\mathbf{72.11}_{0.89}$ | $\mathbf{76.00}_{0.53}$ |

Table 2: Results for out-of-domain settings on ACE 2005 in terms of F1-scores.

## 4.5 Implementation Details

For our method, we use BERT-base as the intervenor and set $\mathcal{N}$ to 100. The constituency parser is from CoreNLP[3]. To ensure that at least one potential relation can be found for each instance, we set $\mathcal{H}$ to 0.8 in all experiments. To ensure quality, we set $\mathcal{K}$ to 1. For a fair comparison, all methods employ GPT-2 base (Radford et al., 2019) as editor.

Following (Zhang et al., 2023), we use two representative PLMs i.e. BERT-base[4] and RoBERTa-base[5] as the base model for RE. To better fit the task, the specific implementation details follow R-BERT (Wu and He, 2019). We use the development set to select the optimal epoch for the base model.

[3] https://stanfordnlp.github.io/CoreNLP/
[4] https://huggingface.co/bert-base-uncased
[5] https://huggingface.co/roberta-base

All reported results are the mean and the standard deviation of micro-F1 value over 5 random seeds. For more implementation details, please check §A.

## 5 Results and Analysis

### 5.1 Results for Low-resource Settings

The results for low-resource settings are shown in Table 1. We make the following observations. 1) CCG achieves the best performance with relatively small standard deviations in almost all cases. The noteworthy point is that the improvement becomes more significant when the data size decreases. For example, the available data decreases from 10% to 1%, CCG gains about 1% to 9% enhancements. 2) The conventional methods exhibit unstable performance, which may lead to negative impacts. 3)

| Method | R-BERT | R-RoBERTa |
|---|---|---|
| Original | $53.34_{1.78}$ | $64.16_{1.19}$ |
| Synonym Rep. | $55.76_{2.07}$ | $65.77_{0.89}$ |
| Back Trans. | $46.61_{1.88}$ | $60.24_{1.20}$ |
| BERT-MLM | $37.85_{0.87}$ | $51.38_{0.70}$ |
| MICE | $62.56_{0.68}$ | $72.77_{0.62}$ |
| AutoCAD | $\underline{63.79}_{0.58}$ | $\underline{73.45}_{0.72}$ |
| CoCo | $59.97_{1.35}$ | $71.29_{0.96}$ |
| ChatGPT | $56.15_{1.18}$ | $65.78_{1.31}$ |
| **CCG** | $\mathbf{70.48}_{1.27}$ | $\mathbf{77.19}_{0.98}$ |

Table 3: Results for adversarial-attack settings on RE-attack in terms of F1-scores. Note that for computational consumption reasons, the augmented data for ChatGPT is obtained by merging data from previous experiments.

| Method | R-BERT | | | |
|---|---|---|---|---|
| | 10% | 32-shot | WL → BC | Adv-attack |
| **CCG** | $\mathbf{78.08}_{0.60}$ | $\mathbf{76.88}_{0.71}$ | $\mathbf{73.16}_{0.60}$ | $\mathbf{70.48}_{1.27}$ |
| *w/o* MII | $76.33_{0.93}$ | $76.14_{1.08}$ | $70.23_{0.88}$ | $68.01_{0.56}$ |
| *w/o* Parser | $\underline{77.63}_{0.74}$ | $\underline{76.87}_{0.44}$ | $\underline{72.48}_{0.61}$ | $\underline{69.36}_{1.55}$ |
| *w/o* LCA | $76.52_{0.55}$ | $75.05_{0.83}$ | $71.17_{1.51}$ | $68.79_{1.65}$ |
| *w/o* WBR | $76.55_{0.78}$ | $73.83_{0.38}$ | $71.64_{1.60}$ | $68.70_{1.10}$ |
| *w/o* Editor | $77.04_{0.82}$ | $72.89_{0.73}$ | $66.50_{1.40}$ | $65.92_{1.22}$ |

Table 4: Results in terms of F1-scores for ablation study, including three settings above mentioned. The best and second best results are in **bold** and underlined, respectively. The subscript denotes the standard deviation.

The counterfactual methods encounter a big challenge under extremely low-resource settings, e.g., 1% and 2-shot scenarios. This is because limited available data prevents the base model from working properly in consistency filtering. 4) Although ChatGPT possesses certain counterfactual reasoning capabilities in low-resource settings, it achieves suboptimal results. After observation, we find that it lacks proficiency in executing relation expansion.

## 5.2 Results for Out-of-domain Settings

Table 2 reports the results for out-of-domain settings. We make the following observations. 1) CCG outperforms all comparative methods in all target domains while maintaining relatively small standard deviations. 2) The conventional methods perform well when data is abundant, e.g., Back Trans. can achieve suboptimal results in WL→BC and WL→NW settings. 3) The enhancement brought by the counterfactual method is quite limited, the reason is that the base model is influenced by in-domain spurious correlations, making it once again unable to perform proper consistency filtering. 4) It is interesting that ChatGPT is always the worst, showing that it is unable to adapt itself in the settings for the counterfactual generation task.

## 5.3 Results for Adversarial-attack Settings

The results for the adversarial-attack settings are shown in Table 3. We make the following observations. 1) CCG achieves the best performance among all methods. The enhancement is quite significant, approximately 13-17%. 2) Generally, the counterfactual methods are superior to the conventional ones, which indicates that counterfactual augmented data provide more reliable assurance in complex and extreme scenarios, driving the base model to lean towards learning causal features.

## 5.4 Ablation Study

To validate the effectiveness of the main components in CCG, we introduce the following variant baselines for the ablation study. 1) CCG *w/o* MII removes the MLM Intervenor-based Indicator (MII) and replaces it with a gradient-based method (Wen et al., 2022). 2) CCG *w/o* Parser removes the constituency parser that is used for alleviating the problem of missing causal words. 3) CCG *w/o* LCA removes the Lowest Common Ancestor (LCA) strategy when discovering potential counterfactual relations. 4) CCG *w/o* WBR removes the WordNet-based Bottom-up Retrieval (WBR) and replaces it with a random selection strategy. 5) CCG *w/o* Editor removes the controlled Editor and directly fills the blank with the most frequent causal term in the dataset corresponding to the target relation. The experimental results are shown in Table 4.

From Table 4, we can observe that, 1) Removing any of the modules or strategies causes performance degradation, which shows that every module in our approach plays an important role. 2) Removing MII or Parser results in up to 2.9% performance degradation, which further emphasizes the importance of identifying causal terms precisely and demonstrates the effectiveness of our approach to improve causal term identification. 3) Removing LCA or WBR results in a 2%-3% decrease, which shows the importance of the hierarchical structure of WordNet for finding target relations that conform to commonsense. 4) Removing Editor results in the most performance degradation, up to 6.3%, as using fixed causal terms introduces new biases.

## 5.5 Human Study

Since it is not possible to automatically evaluate the model's ability of causal terms identification and relation expansion, we conduct a human study.

| **Case 1** | Moby Doll was the first captive `orca` `displayed in` a public aquarium `exhibit` . | *Component-Whole* |
|---|---|---|
| AutoCAD | Moby Doll was the first captive `orca` `caused` `in` a public `exhibit` . ✘ | *Effect-Cause* |
| CoCo | Moby Doll was the primary primary `orca` `generated by` The primary electron `exhibit` . ✘ | *Effect-Cause* |
| ChatGPT | Moby Doll was the first captive `orca` `kept in` a public aquarium `container` . ✘ | *Product-Container* |
| **CCG** | Moby Doll was the first captive `orca` `to` a public aquarium `exhibit` . ✔ | *Entity-Destination* |
| **Case 2** | Colonial `families` of the United States `descended from` the `immigrants` . | *Entity-Origin* |
| AutoCAD | Colonial `families` `discuss` the United States `from` the `immigrants` . ✘ | *Message-Topic* |
| CoCo | Colonial `families` of the United States `enclosed in` the `immigrants` . ✘ | *Content-Container* |
| ChatGPT | Colonial `families` of the United States `belonged to` the `immigrants` . ✘ | *Collection-Member* |
| **CCG** | Colonial `families` of the United States `of` the `immigrants` . ✔ | *Collection-Member* |

Table 5: Abbreviated instances for case study. Entities are in `orange` , causal terms are in `blue` . `Red` denotes missed causal terms, ✔ denotes compliance with the defined requirements, while the ✘ is the opposite.

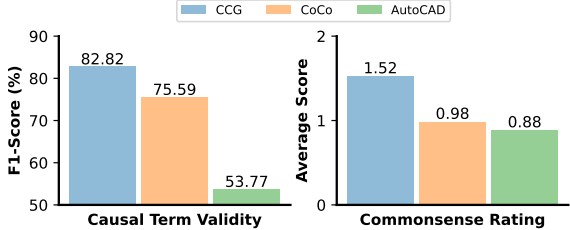

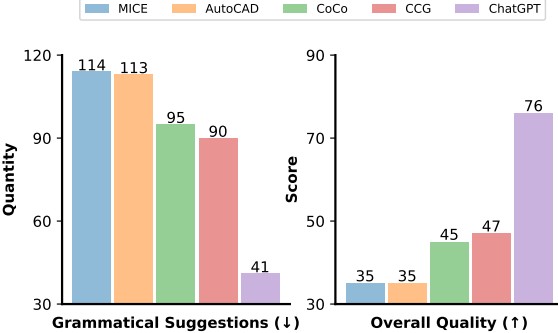

Figure 3: Results of human study. We apply the macro-F1 score for evaluating causal terms identification. To evaluate whether the generated counterfactual conforms to commonsense, we divide the instances into three categories and assign corresponding scores: aligned (2), marginally aligned (1), and not aligned (0).

Specifically, we randomly select 100 samples from SemEval and evaluate counterfactuals generated by CCG, CoCo, and AutoCAD by using manual annotation. The results are presented in Figure 3

We can observe that, 1) CCG substantially outperforms existing methods in finding causal terms, indicating that our proposed causal term identification module aligns best with the human experience. 2) Regarding the evaluation of whether the generated counterfactuals are consistent with commonsense, only CCG scores above 1, i.e., above the level of "marginally aligned". This shows that our proposed commonsense-constrained relation expansion module effectively improves the consistency of counterfactuals with human commonsense, resulting in more reasonable counterfactuals.

### 5.6 Case Study

To have a close look, we randomly select two instances from SemEval for the case study. Table 5 shows the original and augmented instances by CCG and other three typical methods. Clearly, the CCG counterfactuals conform to commonsense and align well with labels. In contrast, the counterfactuals of AutoCAD or CoCo violate the common-

Figure 4: Results of readability study. Grammatical suggestions denote the number of grammatical suggestions by Grammarly, and overall quality denotes the overall quality of writing in this document including readability. ↓ indicates a lower value is better, and ↑ indicates a higher value is better. For example, ChatGPT achieves a score of 76, indicating that the text by ChatGPT is better than 76% of all texts checked by Grammarly.

sense constraint. For example, in Case 1, there is a low likelihood of a *Effect-Cause* relation between entities "orca" and "exhibit". Furthermore, Auto-CAD overlooks partial causal words in Case 1 and Case 2. CoCo is influenced by syntactic parsing errors in Case 1, which reduces the readability of the sentence. ChatGPT either generates illusory relation i.e. non-existent relation "*Product-Container*" in Case 1, or reverses the relation in Case 2.

### 5.7 Readability Study

To thoroughly evaluate the quality of generated counterfactuals, we conduct a readability study to verify grammatical correctness and semantic readability based on Grammarly (Grammarly, 2023). Grammarly is a prevalent English typing assistant that reviews spelling, grammar, punctuation, clarity, engagement, and delivery mistakes in English texts, detects plagiarism, and suggests replacements for the identified errors (Wu et al., 2023).

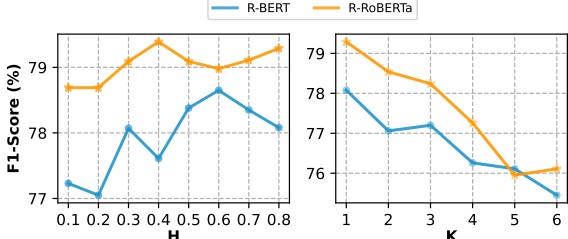

Figure 5: Results of parameter study. We report the results in a 10% setting on the SemEval dataset. $\mathcal{H}$ denotes the ratio threshold to control the scope of upward retrieval in relation expansion. $\mathcal{K}$ denotes the pick-up number in the ranked potential relations, i.e., top $\mathcal{K}$.

Specifically, we randomly select 100 generated counterfactuals on SemEval from each method. We then treat these counterfactuals as a complete document and employ the Grammarly tool to check it. The results are shown in Figure 4. The results demonstrate that the grammatical correctness and semantic readability of CCG are only inferior to ChatGPT, but better than all other compared methods. After all, within an acceptable range of readability, what we focus on more is commonsense.

### 5.8 Parameter Study

The parameters $\mathcal{H}$ and $\mathcal{K}$ affect the amount of data augmentation, and we conducted a parameter study and show the results in Figure 5. Increasing the value of $\mathcal{H}$ indicates an expanded scope for upward retrieval in hypernyms, allowing more entity pairs to uncover potential relations. This continuously improves the model performance within a certain range. However, after reaching a certain threshold, they all start to decline, which validates that augmenting the training set with a small portion leads to more robust models (Treviso et al., 2023).

On the other hand, when more potential relations are taken into account, i.e. increasing the value of $\mathcal{K}$, the performance of the base model shows a continuous downward trend. This phenomenon illustrates that the potential relations discovered earlier are more commonsense, CCG can correctly rank potential relations aligned with commonsense.

### 5.9 Conventional Setting Study

For common datasets, the quantity ratio of the training set to the test set is significantly larger than 1 (e.g. 2.65 in SemEval) and the distribution complies with the IID (Independent Identically Distribution) assumption. To explore the effectiveness of the main counterfactual methods under this set-

| Method | R-BERT | R-RoBERTa |
|---|---|---|
| Original | $88.07_{0.47}$ | $88.22_{0.41}$ |
| MICE | $88.16_{0.25}$ | $87.85_{0.49}$ |
| AutoCAD | $88.18_{0.38}$ | $87.96_{0.52}$ |
| CoCo | $\underline{88.22}_{0.20}$ | $\underline{88.32}_{0.31}$ |
| ChatGPT | $87.52_{1.18}$ | $87.76_{0.26}$ |
| **CCG** | $\mathbf{88.31}_{0.16}$ | $\mathbf{88.45}_{0.38}$ |

Table 6: Results for conventional setting study on SemEval in terms of F1-scores. Numbers in **bold** indicate the best results, and the underlined ones are the second best. The subscript denotes the standard deviation.

ting, we conduct a conventional setting study. The results are presented in Table 6. Consistent with the previous experimental findings (Kaushik et al., 2019; Sen et al., 2021; Wang and Culotta, 2021; Geva et al., 2022), the conventional setting cannot appropriately validate the effectiveness of counterfactuals. The benefits of counterfactuals, from either CCG or CoCo, are quite small under this setting. Other methods even have the opposite effect.

In this situation, the spurious correlations present in the test set are usually contained within the training set. This means that the spurious correlations can assist the model in finding shortcuts and improving accuracy (Sen et al., 2021). Therefore, when counterfactuals block spurious correlations, they may not help the model in terms of accuracy and could even have a counterproductive effect (Kaushik et al., 2019; Sen et al., 2021; Wang and Culotta, 2021; Geva et al., 2022).

### 6 Conclusion

In this paper, we solve the problems in existing methods of CAD, i.e., struggling to accurately identify causal terms under the invariant entity constraint and ignoring the commonsense constraint. We aim to produce the most human-like counterfactuals, i.e., not only grammatically correct and semantically readable but also consistent with commonsense. To this end, we need to satisfy the variant entity and commonsense constraints. To meet the first one, we present a novel intervention-based strategy, which can accurately identify the editable words by potential interventions. To meet the second one, we exploit WordNet to expand potential relations such that the counterfactual generation is constrained by commonsense. Extensive experimental results prove that our framework generates commonsense counterfactuals and outperforms the state-of-the-art baselines. It also consistently enhances the base models' stability across all settings.

## Limitations

Despite the effectiveness of the relation expansion module for mining commonsense relations in CCG, the granularity of WordNet can affect the accuracy of the results, and WordNet as a static knowledge graph also faces the issue of temporal relevance, which makes it challenging for our method to handle emerging concepts. Therefore, we plan to integrate the principles of our algorithm with LLMs at a deep level, as these models encompass rich knowledge. Another limitation of our work is that CCG is still a pipeline framework. Some recent research attempts to jointly optimize a rationale extractor and a classifier in an end-to-end fashion (Paranjape et al., 2020; Yu et al., 2021). We plan to explore this direction.

## Acknowledgments

This work was supported by a grant from the National Natural Science Foundation of China (NSFC) project (No. 62276193). It was also supported by the Joint Laboratory on Credit Science and Technology of CSCI-Wuhan University.

## Ethics Statement

Our work aims to explore the commonsense counterfactual generation, which is entirely at the methodological level and therefore does not have any negative social impact.

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

## A  Experimental Details

### A.1  Datasets

**Low-resource Settings**  Following previous related works (Li et al., 2022a; Chen et al., 2022), we evaluate models' performance under two types of low-resource scenarios. Firstly, Li et al. (2022a) proposed a proportionally divided scenario, including 5% and 10% settings. Namely, randomly sampling 5% or 10% of data from a training set. In addition, we supplement 1% and 3% settings. Secondly, Chen et al. (2022) proposed a scenario divided by a fixed number of instances per relation, including 8-shot, 16-shot, and 32-shot settings. For example, for 8-shot, we randomly extract 8 instances from the training set for each relation. Furthermore, we include 2-shot and 4-shot settings. We introduce these more challenging settings for both scenarios in order to thoroughly assess models' effectiveness.

**Out-of-domain Settings**  ACE 2005 multilingual training corpus (Grishman et al., 2005) contains the complete set of English, Arabic, and Chinese training data for the 2005 Automatic Content Extraction (ACE) technology evaluation. In line with previous work (Zhang et al., 2023), we only conduct experiments on the English data. Due to copyright issues, please download the dataset from the provided link[6]. Refer to the processing approach of previous work (Zhong and Chen, 2021), we utilize the preprocessing code from DyGIE repository[7].

**Adversarial-attack Settings**  To balance the tradeoff between time consumption and data quality, this adversarial-attack dataset is built in a semi-automatic way upon the test set of SemEval (Hendrickx et al., 2019). Specifically, we intervene each instance through label-flipped and label-invariant approaches. In the label-flipped case, for a given instance, we first replace its causal term with the different ones from other relations and flip its label. This allows us to obtain a large number of candidates, but only a small portion of them are readable. Then, we employ GPT-2 (Radford et al., 2019) to calculate their perplexity and select the top 10 sentences with the lowest perplexity for manual verification. Finally, we manually pick at most one sentence that meets the three aforementioned requirements in §3.1 for each instance and apply it to RE-Attack. In the label-invariant situation, for a

---

| Setting | Dataset | Split | Sentences | Types | Direction |
|---------|---------|-------|-----------|-------|-----------|
| Low-resource | SemEval | Train | 7200 | 19 | Bi-direction |
| | | Dev | 800 | | |
| | | Test | 2715 | | |
| Out-of-domain | ACE 2005 | Train (WL) | 576 | 6 | Uni-direction |
| | | Dev (WL) | 192 | | |
| | | Test (BC) | 1607 | | |
| | | Test (BN) | 2015 | | |
| | | Test (NW) | 2680 | | |
| Adversarial-attack | SemEval | Train | 7200 | 19 | Bi-direction |
| | | Dev | 800 | | |
| | RE-Attack | Test | 2391 | | |

Table 7: Statistics of experimental datasets. Be aware that the training set and development set of RE-Attack remain consistent with those of SemEval.

given instance, we first substitute its entities with different entities from instances with the same label. After obtaining various candidates, to pick up high-quality sentences, we also apply the GPT-2-based strategy for automatic filtering. Finally, we manually choose at most one semantically fluent and relation-compliant sentence for each instance from the filtered candidates and apply it to RE-Attack.

### A.2  Baselines

**Conventional Methods**  Following previous work (Wen et al., 2022), the word-replacement ratio of Synonym Replacement (Zhang et al., 2015) and BERT-MLM (Jiao et al., 2019) is set to 30%. Specifically, we randomly select 30% of words in a sentence for replacement. For Synonym Replacement, the synonyms come from WordNet (Miller, 1995). For BERT-MLM, we employ BERT-base[8] as the MLM for word substitution. We implement Back Translation (Sennrich et al., 2015) by using an online translation API[9]. Specifically, we first translate English sentences into Chinese text and then translate them back. To be consistent with previous work (Wen et al., 2022), all the methods mentioned above augment each instance once.

**Counterfactual Methods**  The current counterfactual methods lack a specialized design to conform with the commonsense constraint, including MICE (Ross et al., 2021), AutoCAD (Wen et al., 2022), and CoCo (Zhang et al., 2023). Therefore, during the implementation, these methods assume that all other relations can be potential ones. The ultimate data augmentation in quality control relies entirely on consistency filtering. Simultaneously, the quantity and ratio of data augmentation also depend on the filtering model. The detailed statistics of data augmentation are displayed in Table 8.

---

[6] https://catalog.ldc.upenn.edu/LDC2006T06
[7] https://github.com/luanyi/DyGIE/tree/master/preprocessing

[8] https://huggingface.co/bert-base-uncased
[9] https://api.fanyi.baidu.com/

| Method | Low-resource 1 | | Low-resource 2 | | RE-Attack | | ACE 2005 | |
|---|---|---|---|---|---|---|---|---|
| | Num. | Ratio | Num. | Ratio | Num. | Ratio | Num. | Ratio |
| MICE | 541 | 111% | 282 | 67% | 12894 | 179% | 7 | 1% |
| AutoCAD | 712 | 141% | 368 | 87% | 16839 | 234% | 3 | 1% |
| CoCo | 40 | 8% | 17 | 4% | 677 | 9% | 22 | 4% |
| **CCG** | 266 | 51% | 114 | 54% | 4619 | 64% | 268 | 47% |

Table 8: The statistics of the augmentation quantity and ratio for counterfactual methods. The num. indicates augmentation quantity and the ratio is the proportion of augmentation quantity to the original data volume. Note that the values are averages under two types of low-resource settings, CCG does not require consistency filtering.

**ChatGPT** In order to thoroughly unleash the counterfactual reasoning capability of ChatGPT, we design a clearly described and example-guided prompt, as described in Table 9. In this prompt, we incorporate the Chain-of-thought (COT) for step-wise decomposition, which has been proven effective in reasoning and commonsense to solve tasks (Huang et al., 2022; Wei et al., 2022). We divide the counterfactual generation of RE into three steps and provide specific descriptions for each one. Furthermore, we provide an example to help ChatGPT understand the assignment and standard format. In the implementation, we make use of the official API[10] provided by OpenAI to perform all of our experiments. Specifically, taking into account both performance and resource consumption, the specific version we have chosen is gpt-3.5-turbo. Align with the conventional methods, ChatGPT also augments each instance once.

### A.3 Human Study

To intuitively analyze different strategies, we conduct a small-scale human study. Firstly, to evaluate the performance of various causal term identification strategies, we randomly select 100 instances from SemEval (Hendrickx et al., 2019) for manual annotation, where we manually identify the words that determine the relations between entities. Since we need to adhere to the principle of minimal edits, i.e. identifying causal words (recall) while avoiding affecting non-causal words (precision), we naturally utilize the F1-score for evaluation. It is worth noting that each instance is the smallest unit of data augmentation, hence each instance should be assigned the same weight. Treating each instance as an evaluation category, we employ the Macro-F1 score as a specific evaluation metric. Secondly, to directly evaluate the commonsense degree of generated counterfactuals, from SemEval, we randomly select 100 counterfac-

tual instances generated by each method for rating, respectively. Specifically, we assess an instance whether aligns with commonsense based on the relation role of entities. For instance, entity "buckets" can take on the role of "destination", but not as "effect", as shown in Figure 1. Note that all annotations were completed by three individuals, who possess relevant research experiences, and the final decisions were determined through voting.

---

[10]https://platform.openai.com/

**Task Definition**: changing the relation between entity pair with minimal contextual edits.

**Instruction Description**: the process can be divided into the following three steps. (1) causal term identification, finding as few contextual words (except for entity pair) as possible that can change the relation when they are replaced; (2) potential relation discovery, picking another commonsense relation from candidate relations; (3) causal term replacement, replacing the causal term with appropriate words to change the original relation to the potential one.

**Candidate Relations**: message-topic, topic-message, entity-destination, destination-entity, content-container, container-content, effect-cause, cause-effect, whole-component, component-whole, collection-member, member-collection, agency-instrument, instrument-agency, producer-product, product-producer, entity-origin, origin-entity.

**Example**:
**Input**: the key is moved into a chest.
**Entity Pair**: key-chest
**Relation**: entity-destination
**Causal Term Identification**: relation "entity-destination" depends on contextual words "moved into".
**Potential Relation Discovery**: "key" can be "entity", "chest" can be "origin", thus another commonsense relation from candidate relations is "entity-origin".
**Causal Term Replacement**: to change original relation to the potential one "entity-origin", contextual words "moved into" are replaced with "from".
**Output**: the key is from a chest.
**Relation**: entity-origin

**Inference** (completing the remaining content and maintaining consistency with the format of example):
**Input**: eggs were removed into buckets.
**Entity Pair**: eggs-buckets
**Relation**: entity-destination

Table 9: The prompt designed for directing ChatGPT to generate RE counterfactuals.