# OpenReview forum: "Generating Commonsense Counterfactuals for Stable Relation Extraction"
_EMNLP/2023/Conference — EMNLP 2023 Main_

### Official Review · Reviewer_Ck7v · 2023-07-27

**Soundness:** 4

**Excitement:**

3: Ambivalent: It has merits (e.g., it reports state-of-the-art results, the idea is nice), but there are key weaknesses (e.g., it describes incremental work), and it can significantly benefit from another round of revision. However, I won't object to accepting it if my co-reviewers champion it.

**Missing References:**

"Improving commonsense causal reasoning by adversarial training and data augmentation" by I Staliunaite, PJ Gorinski, I Iacobacci (2021) have presented a very similar method of generating confounders as well as generating adversarial examples to improve commonsense causal reasoning models.

**Paper Topic And Main Contributions:**

The paper presents a novel approach to relation extraction by leveraging commonsense counterfactuals. The authors generate counterfactuals with the view to uncovering relationships between entities. They conduct extensive evaluations and demonstrate that the methods improves the models’ robustness.

**Questions For The Authors:**

Line 141: What does “its performance is entangled with the base model” mean?

Line 290: Why is it obvious? Perhaps it could be useful to briefly state it.

**Reasons To Accept:**

A good overview of related work and clearly delineated unsolved issues and concrete contributions of this paper.

An interesting application of counterfactual generation for relation extraction.

**Reasons To Reject:**

The paper is rather difficult to follow as it lacks structure. Most importantly, terms should be more clearly defined before being used.

**Reproducibility:**

3: Could reproduce the results with some difficulty. The settings of parameters are underspecified or subjectively determined; the training/evaluation data are not widely available.

**Reviewer Confidence:**

3: Pretty sure, but there's a chance I missed something. Although I have a good feel for this area in general, I did not carefully check the paper's details, e.g., the math, experimental design, or novelty.

**Typos Grammar Style And Presentation Improvements:**

Line 001: The start of the abstract is very unclear, the task that the paper tackles should be introduced at the very start, before stating that there are problems in it that most NLP models don’t address.

Line 027: The writing could use with some restructuring, many sentences end with a subclause that makes a side note, which is confusing to read.

Line 070: Invariant entity constraint and commonsense constraint should be clearly defined.

---

> ### Author Rebuttal · Authors · 2023-08-28
>
> We thank the reviewer for the constructive comments. We hope the following clarifications can address the reviewer’s concerns.
>
> ***Question:** Line 141: What does “its performance is entangled with the base model” mean?*
> **Response:** As mentioned in Line 060, current counterfactual augmentation methods rely on filtering strategies to ensure the quality of the generated data. In this process, the filter is the relation extraction model (the base model) trained on existing data, and the data the base model filters is used to improve itself. We define this contradictory phenomenon as the entangled problem. Note that our method identifies explicit targets through potential relation identification, which enables the direct generation of high-quality counterfactuals. As a result, this process can be bypassed, i.e., we effectively address this issue at its core.
>
> ***Question:** Line 290: Why is it obvious? Perhaps it could be useful to briefly state it.*
> **Response:** As introduced in Line 260, hypernymy (super-name) are transitive relations between concepts. Moreover, concepts gradually become broader in the hierarchy of hypernyms, as illustrated in Figure 2 in our submission (bin->container->artifact, the container is hypernymy of the bin, the artifact is hypernymy of the container). Therefore, between any entity pairs, the lower common hypernym in the hierarchical structure, the closer their semantics. According to this nature, for a given entity pair, our bottom-up retrieval strategy can proactively identify semantically closer entity pairs with different relations. As a result, the relations discovered earlier come from the entity pairs with more similar semantics, hence it should be given a higher priority. We will add these statements to the paper per your suggestion.
>
> ***Comment:** "Improving commonsense causal reasoning by adversarial training and data augmentation" by I Staliunaite, PJ Gorinski, I Iacobacci (2021) have presented a very similar method of generating confounders as well as generating adversarial examples to improve commonsense causal reasoning models.*
> **Response:** Regarding the reference you introduced, we respectfully disagree with your mention of "a very similar method of generating confounders as well as generating adversarial examples". Our reasons are as follows.
> 1) There is an essential distinction between adversarial examples and counterfactual samples. The methods in the reference utilize synonym substitution to generate adversarial samples, whereas our focus is on generating counterfactual samples. From a causal perspective, the synonym substitution approach belongs to the second level of the causal ladder, namely intervention. It does not require a causal discovery process and relies on the principle of semantic invariance, and implementing the intervention (with unchanged labels) through the simple synonym substitution. In simpler terms, it guides the model by saying "you shouldn't be affected by irrelevant perturbations". In contrast, counterfactuals belong to the third level of the causal ladder, where they require causal discovery methods to identify causal words and then intervene on them to emphasize decision boundaries (label flipping). In simpler terms, the counterfactual-based approach directly guides the model by saying "you should make judgments based on causal-relevant information". Therefore, our method does not generate confounders. Moreover, we have taken a step further by first addressing and validating the significance of commonsense in counterfactual generation, which is evidently different from the commonsense classification task that the reference has focused on. Additionally, the adversarial-attack setting we employed is just one way to validate the effectiveness of counterfactuals, rather than directly attacking the target model.
> &nbsp;
> 2) Due to the distinction of the problems, there naturally exist significant differences in the implementation of methods.
> a) Firstly, the reference method does not require a causal discovery process, as mentioned earlier.
> b) Secondly, the reference method solely relies on the node information from WordNet while our method leverages both the node and edge information. WordNet is a knowledge graph based on human commonsense, consisting of the synsets (conceptual nodes) and their relations (edges like hypernyms or hyponyms). Our method not only utilizes the nodes for concept localization, but also leverages hypernym relations between concepts to generalize entities. This enables the propagation of relations, thereby uncovering potential relations aligned with commonsense, as detailed in Algorithm 1.
> c) Thirdly, the data generation process of the reference involves only substitution and does not involve generative models. On the contrary, generating counterfactual examples involves controlled text generation, which is also a challenge. After obtaining potential relations, the generative model needs to produce causal words consistent with the new relations.
> d) Finally, the counterfactual data we generate can be directly used for data augmentation to enhance model stability. This process eliminates the need for the attack procedure relied upon by the reference, resembling the entangled issue we mentioned earlier.
> &nbsp;
> 3) Accompanied by the challenge during the counterfactual generation, counterfactuals are more effective in eliminating spurious correlations. The reference method is similar to the synonym substitution method Synonym Rep. which we have used as a baseline. The comparative analysis of the results between Synonym Rep. and our CCG indicates that synonym substitution data eliminates spurious correlations through a process of exclusion, whereas counterfactuals clearly emphasize the decision boundary directly, which proves to be more efficient.
>
> ***Suggestion:** Line 001: The start of the abstract is very unclear, the task that the paper tackles should be introduced at the very start, before stating that there are problems in it that most NLP models don’t address.
> Line 027: The writing could use with some restructuring, many sentences end with a subclause that makes a side note, which is confusing to read.
> Line 070: Invariant entity constraint and commonsense constraint should be clearly defined.*
> **Response:** So many thanks for your intensive suggestions, we will make adjustments to the abstract and the main text accordingly.

---

### Official Review · Reviewer_Uqod · 2023-08-03

**Typos Grammar Style And Presentation Improvements:** 1.	Missing details
**Soundness:** 4

**Excitement:**

3: Ambivalent: It has merits (e.g., it reports state-of-the-art results, the idea is nice), but there are key weaknesses (e.g., it describes incremental work), and it can significantly benefit from another round of revision. However, I won't object to accepting it if my co-reviewers champion it.

**Paper Topic And Main Contributions:**

This paper introduces Commonsense Counterfactual Generation (CCG), a new counterfactual data augmentation method for relation extraction. It tries to resolve two challenges: the first is to accurately identify causal terms, and the second is to be consistent with commonsense. The method is divided into three steps, causal terms identification, relation expansion, and controlled editing. In causal terms identification, the authors design an intervention-based strategy to identify editable words and use a constituency parser for correction. In relation expansion, they uncover possible relations that meet commonsense with the help of WordNet. In controlled editing, a content generation model is trained to generate the masked content given a relation. Experiments in various settings demonstrate the effectiveness and robustness of CCG.

**Questions For The Authors:**

The ChatGPT prompt divides the task into three steps. I wonder in what steps ChatGPT does not perform well. A defect shown in the case study is that ChatGPT may generate illusory relation in potential relation discovery. If this is the main problem, can it be easily solved by filtering?

**Reasons To Accept:**

1.	The paper is well organized. It clearly defines the requirements of counterfactual data augmentation for relation extraction, and summarizes two reasonable challenges for this task. The proposed method fits the motivations nicely.
2.	The authors conduct multidimensional experiments and analysis. They explore three different scenarios: low-resource, out-of-domain, and adversarial-attack, and CCG exhibits its advantage in all the scenarios.

**Reasons To Reject:**

1.	CCG performs well when reading the numbers in tables, but it doesn't appear to be that good when seeing the generated outputs in case study. In both randomly selected cases, it replaces verb phrase with a single preposition, which undermines the integrity of the sentence, while other methods generate grammatically correct sentences. A human evaluation on the grammatical correctness and semantical readability is possibly needed to address the problem.
2.	The experiments focus on a small amount of training data, and as the experiments on SemEval show, the performance gain of CCG diminishes when the training data increases. When the training data comes to 10% and 32-shot, the difference between CCG and CoCo 	is smaller than the standard deviation. I find that CoCo, the only existing research towards this task, experiments on some large-scale settings, like the whole SemEval training set and all domains except one in the out-of-domain setting. So adding experiments following the setting of CoCo may help to compare the two methods comprehensively.

**Reproducibility:**

4: Could mostly reproduce the results, but there may be some variation because of sample variance or minor variations in their interpretation of the protocol or method.

**Reviewer Confidence:**

4: Quite sure. I tried to check the important points carefully. It's unlikely, though conceivable, that I missed something that should affect my ratings.

---

> ### Author Rebuttal · Authors · 2023-08-28
>
> Thank you for your valuable suggestions. Based on your advice and questions, we have made the following clarifications in an effort to address your concerns.
>
> ***Suggestion:** CCG performs well when reading the numbers in tables, but it doesn't appear to be that good when seeing the generated outputs in case study. In both randomly selected cases, it replaces verb phrase with a single preposition, which undermines the integrity of the sentence, while other methods generate grammatically correct sentences. A human evaluation on the grammatical correctness and semantical readability is possibly needed to address the problem.*
> **Response:** Regarding the grammatical correctness and semantical readability problem, the first case by our CCG is correct since we replace the past participle with a preposition, and the readability of the second case by CCG, while satisfying the commonsense constraint, is not as good as those by CoCo and ChatGPT.
> &emsp; Please note that other methods also generate grammatically incorrect and semantical unreadable sentences, as shown in two cases. To thoroughly evaluate the quality of generated counterfactuals, we conduct evaluation experiments to verify grammatical correctness and semantic readability based on Grammarly [1]. Grammarly is a prevalent English typing assistant that reviews spelling, grammar, punctuation, clarity, engagement, and delivery mistakes in English texts, detects plagiarism and suggests replacements for the identified errors [2]. Specifically, we randomly select 100 generated examples from each counterfactual-based methods. We then treat these examples as a complete document and let the Grammarly tool check it. The results are shown in Table 1.
>
> | Method | Suggestion (↓) | Score (↑) |
> | --- | --- | --- |
> | MICE | 114 | 35 |
> | AutoCAD | 113 | 35 |
> | CoCo | 95 | 45 |
> | ChatGPT | $\textbf{41}$ | $\textbf{76}$ |
> | CCG | $\underline{90}$ | $\underline{47}$ |
> |  |  |  |
>
> Table 1: Results for grammatical correctness and semantic readability evaluation on SemEval, where ‘*suggestion*’ denotes the number of grammatical suggestions by Grammarly, and ‘*score*’ denotes the overall quality of writing in this document including readability. ↓ indicates that a lower value is better, and ↑ indicates that a higher value is better. Numbers in **bold** indicate the best results, and the $\underline{underlined}$ ones are the second best. For example, ChatGPT achieves a score of 76, indicating that the text by ChatGPT is better than 76% of all texts checked by Grammarly including those by humans.
>
> The results demonstrate that the grammatical correctness and semantic readability of CCG is only inferior to ChatGPT, but better than all other methods. We will further conduct a human evaluation and will report the results on randomly selected 10 instances (due to the space limit) from the above set in the appendix.
>
> ***Suggestion:** The experiments focus on a small amount of training data, and as the experiments on SemEval show, the performance gain of CCG diminishes when the training data increases. When the training data comes to 10% and 32-shot, the difference between CCG and CoCo is smaller than the standard deviation. I find that CoCo, the only existing research on this task, experiments on some large-scale settings, like the whole SemEval training set and all domains except one in the out-of-domain setting. So adding experiments following the setting of CoCo may help to compare the two methods comprehensively.*
> **Response:** According to our experience and the results of previous work [3-6], we believe that the large-scale setting under the IID (Independent Identically Distribution) scenario cannot effectively validate the effects of counterfactuals. Usually, the ratio of a training set to a test set is significantly larger than 1 (e.g. 2.65 in SemEval), and if under an IID scenario, the spurious correlations present in the test set are also contained within the training set. In this situation, the spurious correlations can assist the model in finding shortcuts and improving accuracy [4]. Therefore, when counterfactuals block spurious correlations, they may not help the model in terms of accuracy and could even have a counterproductive effect [3-6].
> &emsp; To eliminate such “benefit” of spurious correlations and accurately validate the effects of counterfactuals, we need to disrupt this contained relationship of spurious correlations between training and test set. For the training set, we reduce the data size to reduce its overlap with the spurious associations present in the test set. This corresponds to the low-resource setting that we introduced (note that the test set remains unchanged in this case). For the test set, we can introduce out-of-domain and out-of-distribution instances to make its distribution different from the training set. These correspond to the our-of-domain and adversarial-attack settings that we introduced. Please note that the training set remains unchanged in this case, e.g., we use the whole SemEval training set in the adversarial-attack setting. As a result, both low-resource and out-of-domain settings have been considered as important ways to validate counterfactuals [7].
> &emsp; Nonetheless, we have reported the F1-socre compared with other counterfactual-based methods on the whole SemEval training set. The results are presented in Table 2. Due to the reasons mentioned above, the benefits of counterfactuals, from either CoCo or CCG, are quite small under this setting. Other methods even have the opposite effect. This proves that such a setting is impropriate to truly showcase the quality differences of counterfactuals.
>
> | Method | R-BERT | R-RoBERTa |
> | --- | --- | --- |
> | Original | 88.07 (± 0.47) | 88.22 (±0.41) |
> | MICE | 88.16 (±0.25) | 87.85 (±0.49) |
> | AutoCAD | 88.18 (±0.38) | 87.96 (±0.52) |
> | CoCo | $\underline{88.22}$ (±0.20) | $\underline{88.32}$ (±0.31) |
> | ChatGPT | 87.52 (±0.29) | 87.76 (±0.26) |
> | CCG | $\textbf{88.31}$ (±0.16) | $\textbf{88.45}$ (±0.38) |
> |  |  |  |
>
> Table 2: The comparison results on the whole SemEval training set. Numbers in **bold** indicate the best result, and the $\underline{underlined}$ ones are the second best. The numbers within parentheses indicate the standard deviation.
>
> In addition, there is one more point we need to clarify. In the out-of-domain setting, our experimental setup is consistent with CoCo, that is, one domain serves as the training set, while the remaining domains are used as separate test sets. The results have been presented in Table 2 of our submitted paper.
>
> ***Suggestion:** The ChatGPT prompt divides the task into three steps. I wonder in what steps ChatGPT does not perform well. A defect shown in the case study is that ChatGPT may generate illusory relations in potential relation discovery. If this is the main problem, can it be easily solved by filtering?*
> **Response:** To analyze which step ChatGPT does not perform well, we conduct a human study. Specifically, we randomly select 100 examples generated by ChatGPT. And then, we count the number of errors for each step and calculate the proportions. The errors in the first, second, and third steps account for 10%, 48%, and 42%, respectively. We can observe that the second step, where potential relations identification, performs the worst, and the third step is also affected. Therefore, generating illusory relations is the main problem.
> &emsp; Afterward, we attempt to apply a filtering mechanism to ChatGPT. We report F1-socre in Table 3 and Table 4. Table 3 includes low-resource and adversarial-attack settings, and Table 4 corresponds to out-of-domain setting.
>
> | Method | | | R-BERT | | | | | R-RoBERTa | | |
> | --- | --- | --- | --- | --- | --- | --- | --- | --- | --- | --- |
> | | 1% | 3% | 5% | 10% | Adv. | 1% | 3% | 5% | 10% | Adv. | 1% | 3% | 5% | 10% | Adv.
> | Original | 33.26 (±1.43) | 59.31 (±1.46) | 68.66 (±1.77) | 76.47 (±1.14) | 53.34 (±1.78) | 35.77 (±2.41) | 64.27 (±3.20) | 69.99 (±1.84) | 78.27 (±1.07) | 64.16 (±1.19) |
> | ChatGPT | 38.78 (±2.71) | 61.84 (±1.23) | 67.90 (±2.14) | 75.15 (±1.10) | 56.15 (±1.18) | 38.71 (±2.11) | 64.44 (±1.34) | 70.14 (±2.11) | 76.25 (±0.52) | 65.78 (±1.31) |
> | w/ Filtering | 32.47 (±2.87) ↓ | 59.85 (±2.06) ↓ | 68.39 (±2.52) ↑ | 78.64 (±0.37) ↑ | 55.70 (±2.30) ↓ | 35.46 (±2.54) ↓ | 65.37 (±1.68) ↑ | 72.41 (±2.78) ↑ | 79.33 (±1.27) ↑ | 64.67 (±1.37) ↓ |
> |  |  |  |  |  |  |  |  |  |  |  |
>
> Table 3: The results of low-resource and adversarial-attack settings. 1%-10% denotes the low-resource setting, and Adv. denotes the adversarial-attack setting. w/ Filtering represents ChatGPT combined with the filtering mechanism. ↓ indicates a decrease compared to the original result, while ↑ indicates an increase.
>
> | Method | | R-BERT | | | R-RoBERTa | |
> | --- | --- | --- | --- | --- | --- | --- |
> | | WL→BC | WL→BN | WL→NW | WL→BC | WL→BN | WL→NW |
> | Original |   70.43 (±2.45) | 70.55 (±2.51) | 69.42 (±1.41) | 74.17 (±0.70) | 70.54 (±0.87) | 74.93 (±0.74) |
> | ChatGPT | 52.70 (±0.99) | 55.94 (±1.21) | 54.51 (±0.63) | 59.55 (±0.50) | 60.11 (±0.89) | 61.74 (±1.13) |
> | w/ Filtering | 69.27 (±2.37) ↑ | 70.46 (±1.93) ↑ | 69.55 (±1.63) ↑ | 74.58 (±1.37) ↑ | 70.33 (±1.36) ↑ | 74.85 (±0.74) ↑ |
> |  |  |  |  |  |  |  |
>
> Table 4: The results of out-of-domain setting. WL→BC denotes that the training set is in the WL domain and the test set is in the BC domain, the same for others. w/ Filtering represents ChatGPT combined with the filtering mechanism. ↓ indicates a decrease compared to the original result, while ↑ indicates an increase.
>
> Based on the experimental results, we can conclude that the problem of illusory relation can be partially alleviated through filtering in some cases, but it cannot be easily solved.
> &emsp; Firstly, although the filtering mechanism can remove noise data, it heavily relies on the performance of the filter, i.e., the base model. In settings with relatively abundant training resources, the filter is adequately trained and the filtering mechanism might be effective, such as the cases in the 5%, 10%, and the out-of-domain settings. However, in more extreme scenarios such as very low-resource or adversarial-attack settings, the filter itself may become ineffective.
> &emsp; Secondly, the filtering mechanism, which optimizes data solely through subtraction, always has its limitations. Filtering only mitigates the negative impact of low-quality data without generating higher-quality data. Therefore, even with the inclusion of the filtering mechanism, the vast majority of results still exhibit significant differences from CCG.
>
> ***Question:** Line 97: what does “they” refer to?
> Line 230: why determining the change of prediction is much simpler than predicting the actual outcomes?
> Algorithm 1: the hyperparameter K is missing.*
> **Response:** In Line 97, “They” refers to the current counterfactual generation methods in the natural language processing community, including MICE, AutoCAD, and CoCo.
> In Line 230, the decision space for “determining the change of prediction” (binary classification) is much smaller than the decision space for“predicting the actual outcomes” (multi-class classification, directly proportional to the number of relations), which greatly reduces the complexity of the problem.
> Thank you for your reminder. We will incorporate the hyperparameter K into the Algorithm 1.
>
> ***Suggestion:** Line 214: an intervention-based strategy
> Line 369: The constituency parser is from CoreNLP.
> Caption of Table 7: The instances are from SemEval dataset.*
> **Response:** So many thanks for your intensive reading! We will carefully edit the paper and make revisions per your suggestions!
>
> **Rerefrence**
> [1] Grammarly. 2023.https://www.grammarly.com/about.
> [2] Wu H, Wang W, Wan Y, et al. Chatgpt or grammarly? evaluating chatgpt on grammatical error correction benchmark[J]. arXiv preprint arXiv:2303.13648, 2023.
> [3] Kaushik D, Hovy E, Lipton Z C. Learning the difference that makes a difference with counterfactually-augmented data[J]. arXiv preprint arXiv:1909.12434, 2019.
> [4] Sen I, Samory M, Flöck F, et al. How Does Counterfactually Augmented Data Impact Models for Social Computing Constructs?[J]. arXiv preprint arXiv:2109.07022, 2021.
> [5] Wang Z, Culotta A. Robustness to spurious correlations in text classification via automatically generated counterfactuals[C]//Proceedings of the AAAI Conference on Artificial Intelligence. 2021, 35(16): 14024-14031.
> [6] Geva M, Wolfson T, Berant J. Break, perturb, build: Automatic perturbation of reasoning paths through question decomposition[J]. Transactions of the Association for Computational Linguistics, 2022, 10: 111-126.
> [7] Calderon N, Ben-David E, Feder A, et al. Docogen: Domain counterfactual generation for low resource domain adaptation[J]. arXiv preprint arXiv:2202.12350, 2022.

---

### Official Review · Reviewer_fvyK · 2023-08-07

**Soundness:** 4

**Excitement:**

4: Strong: This paper deepens the understanding of some phenomenon or lowers the barriers to an existing research direction.

**Paper Topic And Main Contributions:**

This work proposes to use counterfactual augmented data to improve relation extraction methods. Concretely authors aim to use counterfactuals to avoid relying on spurious correlations.
The method relies on three aspects. To generate counterfactual data only affects the context, to keep the entity pair while changing the relation entity. The method identifies the causal term, and removes the minimal phrase of it. Identifies hypernyms of the participating entities and their relation to do relation expansion.
The model is evaluated in three different settings: low resource setting, out of domain and adversarial setting. For the last one, a new dataset “REAttack” was introduced.
Model is compared with reasonable baselines such as synonym replacement, back translation BERT-MLM, MICE (Ross et al., 2021) AutoCAD (Wen et al., 2022), CoCo (Zhang et al., 2023) and ChatGPT.
Numbers show that the proposed model Commonsense Counterfactual Generation (CCG) presents better behavior than its competitors.
A human evaluation is also performed in order to evaluate the quality of the counterfactuals of CCG compared to CoCo and AutoCAD, showing a clear differentiation in terms of F1 for causal term validity and score for commonsense rating


**Reasons To Accept:**

Approach uses a known method (counterfactual augmented data) on an important NLP task (relation extraction).

Proposal seems fairly grounded by previous work (authors did a good job in exploring previous work, locating this work in the research space and providing clear motivations).

Experimentation was carried out on reasonable datasets. ACE2005 is standard in relation extraction domain.

Experiments show an improvement in performance compared to previous work.


**Reasons To Reject:**

Solid work… no clear reasons to reject


**Reproducibility:**

5: Could easily reproduce the results.

**Reviewer Confidence:**

3: Pretty sure, but there's a chance I missed something. Although I have a good feel for this area in general, I did not carefully check the paper's details, e.g., the math, experimental design, or novelty.

**Typos Grammar Style And Presentation Improvements:**

In general: sort citations by date i.e.  (Wang and Culotta, 2021; Garg and Ramakrishnan, 2020; Kaushik et al., 2019) => Kaushik et al., 2019 Garg and Ramakrishnan, 2020, Wang and Culotta, 2021)

Conclusions seem a bit misaligned with abstract. Authors note in the abstract that ”existing methods [of counterfactual augmented data] encounter [..] problems when dealing with the fine-grained relation extraction tasks”. In the conclusions authors claim that they “introduce the problem of commonsense counterfactual generation into the relation extraction field”, which is not the same.
Also, “commonsense counterfactual generation” is more a technique rather than an problem. The problem, as stated in the abstract is the weak quality of that generation due to: “struggle to accurately identify causal term under the invariant entity constraint” and “ignore the commonsense constraint”

---

> ### Author Rebuttal · Authors · 2023-08-28
>
> Thank you very much for your valuable suggestions. We will take them into consideration and make the necessary modifications to the paper.
>
> ***Suggestion:** sort citations by date i.e. (Wang and Culotta, 2021; Garg and Ramakrishnan, 2020; Kaushik et al., 2019) => Kaushik et al., 2019 Garg and Ramakrishnan, 2020, Wang and Culotta, 2021)*
> **Response:** So many thanks for your intensive reading! We will carefully edit the paper and sort all citations by date per your advice.
>
> ***Suggestion:** Conclusions seem a bit misaligned with abstract. Authors note in the abstract that ”existing methods [of counterfactual augmented data] encounter [..] problems when dealing with the fine-grained relation extraction tasks”. In the conclusions authors claim that they “introduce the problem of commonsense counterfactual generation into the relation extraction field”, which is not the same. Also, “commonsense counterfactual generation” is more a technique rather than an problem. The problem, as stated in the abstract is the weak quality of that generation due to: “struggle to accurately identify causal term under the invariant entity constraint” and “ignore the commonsense constraint”.*
> **Response:** Thank you for pointing this out! We will revise the conclusion part as follows: “… solve the problems in existing methods of counterfactual generation, i.e., struggling to accurately identify causal term under the invariant entity constraint and ignoring the commonsense constraint.”

---

### Meta-Review · Area_Chair_zKxU · 2023-09-19

**Recommendation:** 3

**Metareview:**

This paper presents a novel approach to data augmentation for relation extraction, consisting of three key steps: causal term identification, relation expansion, and controlled editing. Generally speaking, this is a solid data augmentation work for relation extraction. While the paper employs terms such as "intervention," "counterfactuals," and "causal," it is my viewpoint that the method proposed in this paper has a relatively weak connection to causal theory and does not operate within a causal theory framework. Therefore, I recommend that the authors carefully reconsider this aspect, make necessary revisions to the paper, and provide a clearer elucidation of the relationship between their method and causal theory, or employ more appropriate descriptions.

---

### Decision · Program_Chairs · 2023-10-07

**Decision:**

Accept-Main

**Comment:**

This paper presents a novel approach to data augmentation for relation extraction, consisting of three key steps: causal term identification, relation expansion, and controlled editing. Generally speaking, this is a solid data augmentation work for relation extraction. While the paper employs terms such as "intervention," "counterfactuals," and "causal," it is my viewpoint that the method proposed in this paper has a relatively weak connection to causal theory and does not operate within a causal theory framework. Therefore, I recommend that the authors carefully reconsider this aspect, make necessary revisions to the paper, and provide a clearer elucidation of the relationship between their method and causal theory, or employ more appropriate descriptions.